# Sex Differences in Lipid Profile across the Life Span in Patients with Type 2 Diabetes: A Primary Care-Based Study

**DOI:** 10.3390/jcm10081775

**Published:** 2021-04-19

**Authors:** Martina Ambrož, Sieta T. de Vries, Priya Vart, Robin P. F. Dullaart, Jeanine Roeters van Lennep, Petra Denig, Klaas Hoogenberg

**Affiliations:** 1Department of Clinical Pharmacy and Pharmacology, University of Groningen, University Medical Center Groningen, 9700RB Groningen, The Netherlands; s.t.de.vries@umcg.nl (S.T.d.V.); p.vart@umcg.nl (P.V.); p.denig@umcg.nl (P.D.); 2Department of Internal Medicine-Endocrinology, University of Groningen, University Medical Center Groningen, 9700RB Groningen, The Netherlands; r.p.f.dullaart@umcg.nl; 3Department of Internal Medicine, Erasmus Medical Center, University Medical Center Rotterdam, 3015GD Rotterdam, The Netherlands; j.roetersvanlennep@erasmusmc.nl; 4Department of Internal Medicine, Martini Hospital, 9728NT Groningen, The Netherlands; hoogenk@mzh.nl

**Keywords:** sex differences, age-related, life span, lipids, cholesterol, diabetes mellitus, type 2, statins, primary care

## Abstract

We assessed sex differences across the life span in the lipid profile of type 2 diabetes (T2D) patients treated and not treated with statins. We used the Groningen Initiative to ANalyze Type 2 diabetes Treatment database, which includes T2D patients from the north of the Netherlands. Patients with a full lipid profile determined between 2010 and 2012 were included. We excluded patients treated with other lipid-lowering drugs than statins. Sex differences in low- and high-density lipoprotein cholesterol (LDL-c and HDL-c) and triglyceride (TG) levels across 11 age groups stratified by statin treatment were assessed using linear regression. We included 26,849 patients (51% women, 55% treated with statins). Without statins, women had significantly lower LDL-c levels than men before the age of 45 years, similar levels between 45 and 49 years, and higher levels thereafter. With statins, similar LDL-c levels were shown up to the age of 55, and higher levels in women thereafter. Women had significantly higher HDL-c levels than men, regardless of age or statin treatment. Men had significantly higher TG levels up to the age of 55 and 60, depending on whether they did not take or took statins, respectively, and similar levels thereafter. When managing cardiovascular risk in patients with T2D, attention is needed for the menopausal status of women and for TG levels in younger men.

## 1. Introduction

Cardiovascular disease (CVD) is the leading cause of mortality in the world [1], and a person’s lipid profile is an important aspect of the cardiovascular risk. It has been shown that men have a higher risk of atherosclerotic CVD than women [2] and that men develop CVD on average seven to ten years earlier than women [3]. Women are assumed to have more cardiometabolic reserves associated with female sex hormones, which gives them a biologic advantage when it comes to cardiovascular risk [4,5,6]. At younger ages, women have a more favorable lipid profile, characterized by lower levels of low-density lipoprotein cholesterol (LDL-c) and higher levels of high-density lipoprotein cholesterol (HDL-c) than men [7]. During the menopausal transition, women develop a more adverse lipid profile, characterized by an increase in LDL-c and a decrease in HDL-c [8,9,10,11].

In patients with type 2 diabetes (T2D), the risk of CVD is at least doubled in comparison to patients without diabetes [12,13]. Although T2D is a risk factor for both men and women, the impact of T2D on cardiovascular risk is markedly higher in women. Recent meta-analyses showed that T2D poses a 44% greater excess risk for coronary heart disease and a 27% greater excess risk for stroke in women compared to men [14,15]. Sex differences in the lipid profile are likely to play a role in this, since women with T2D have higher LDL-c and HDL-c levels and lower triglyceride (TG) levels than men with T2D [16]. It is not clear, however, whether such differences are present across all age groups and if they are influenced by menopausal status. This is likely, since a more atherogenic lipid and pro-inflammatory profile has been shown in postmenopausal diabetic women when compared to premenopausal non-diabetic or diabetic women [17].

To reduce cardiovascular risk in patients with T2D, a lipid-lowering treatment with a statin has been recommended for most of these patients, without differentiating between men and women [18,19]. The Cholesterol Treatment Trialists’ Collaborators have shown that the proportional reduction in major vascular events per mmol/L of LDL-c reduction with statins is similar between men and women with T2D [20]. Since T2D increases the risk of CVD in women more than in men, women should be treated at least as stringently as men [6]. Nonetheless, it seems that women with T2D are treated less aggressively with statins [21,22,23] and achieve cholesterol treatment targets less often than men [16,21,22,24,25,26,27].

Taken together, the above highlighted findings suggest that not only the presence of T2D but also menopausal status is relevant for the observed sex differences in cardiovascular risk. So far, a comprehensive analysis of sex differences in the lipid profile across the life span in patients with T2D is lacking. Furthermore, it is not clear to what extent possible sex differences across age groups can be mitigated by treatment with statins. Therefore, the aim of this study was to assess differences in the lipid profile between men and women with T2D across the life span and to assess to what extent are such differences influenced by treatment with statins. This information can provide insight into potentially undertreated populations and help guide personalized treatment.

## 2. Materials and Methods

### 2.1. Study Design and Population

We conducted a cross-sectional cohort study using data from the Groningen Initiative to ANalyse Type-2 diabetes Treatment (see GIANTT at https://umcgresearch.org/facilities, accessed on 7 April 2021) database. This database contains anonymous primary care electronic medical record data from patients with T2D in the northern part of the Netherlands, including outcomes of diagnostic measurements and medication prescriptions. This population mostly consists of Caucasian people. The GIANTT data have been used for numerous studies [23,28,29], including a study in which GIANTT was the reference care-as-usual cohort [30]. Data imports are checked for completeness, and measurement units, coding of medication, and diagnostic measurements are harmonized before being imported in GIANTT.

Patients were included if they were treated by a general practitioner, had at least one full lipid profile (i.e., total cholesterol (TC), LDL-c, HDL-c, and TG) measurement between 1 January 2010 and 31 December 2012, had information about medical history of at least 180 days before the date of the lipid profile measurement, and were aged 18 years or older. We excluded patients without a known date of T2D diagnosis and those treated with other lipid-lowering drugs than statins (i.e., fibrates, bile acid sequestrants, nicotinic acid and derivatives, other lipid-modifying agents, or statins in a combination with other lipid-lowering drugs). The first date of the full lipid profile measurement was defined as the index date.

We obtained an exemption letter from the University Medical Center Groningen Medical Ethics Review Board (reference number M20.257509) indicating that an approval from the ethics committee was not needed for this study using anonymous data in the Netherlands.

### 2.2. Outcome Variables

Our primary outcomes were the LDL-c, HDL-c, and TG levels in mmol/L at the index date. The secondary outcomes were the levels of TC and non-HDL cholesterol (non-HDL-c) at the index date.

Total cholesterol, LDL-c, HDL-c, and TG levels were assessed directly with standard enzymatic colorimetric methods (Roche elecsys C Module; Roche diagnostics, Switzerland) after an overnight fast. Non-HDL-c was calculated by subtracting HDL-c from TC.

### 2.3. Explanatory Variables

Sex and age were included as explanatory variables in our analyses. Sex was used as registered in GIANTT and defined as man or woman. Age was calculated on the index date and categorized in 11 age groups: <40 years, 40–44 years, 45–49 years, 50–54 years, 55–59 years, 60–64 years, 65–69 years, 70–74 years, 75–79 years, 80–84 years, and ≥85 years.

### 2.4. Confounders

We considered including body mass index (BMI), glycated hemoglobin A1c (HbA1c), and smoking status (smoker vs. non-smoker) as possible confounders. BMI and HbA1c had less than 20% of missing values, which were imputed using multiple imputation by chained equations (MICE; Appendix A). Smoking status was missing for more than 60% of patients and was therefore not included in our analyses.

### 2.5. Analyses

Patient characteristics per treatment group, sex, and age group were analyzed descriptively. More information about the time periods and definitions used for the patient characteristics can be found in Appendix A.

We conducted linear regression analyses to assess differences in lipid levels between men and women across different age groups, including an interaction term between sex and age groups. BMI (continuous) was included as a possible confounder for all outcomes, since BMI differed between sex and age groups and it has been associated with the lipid profile. HbA1c (continuous) was included as a possible confounder in the analysis of TG due to its relationship with TG levels [31]. Adjusted mean lipid levels with their 95% confidence intervals were estimated for all sex and age groups. The analyses were conducted separately for patients treated and not treated with a statin. Statin treatment was defined as the prescription of a statin in at least two out of three months before the index date. All statins prescribed and available in the study period were included, i.e., simvastatin, pravastatin, fluvastatin, atorvastatin, and rosuvastatin.

Sensitivity analyses were conducted for the primary outcome among those treated with a statin in which we additionally adjusted for moderate-intensity treatment (i.e., simvastatin, pravastatin, fluvastatin, atorvastatin < 40 mg, and rosuvastatin < 20 mg) versus high-intensity treatment (i.e., atorvastatin ≥ 40 mg and rosuvastatin ≥ 20 mg) [18] with a statin (binary variable).

All analyses were conducted in Stata version 14 (Stata Corp., College Station, TX, USA), and two-sided *p*-values < 0.05 were considered statistically significant.

## 3. Results

There were 26,849 patients included in this study (Figure 1), of which 13,733 (51%) were women, and 14,894 (55%) were treated with a statin. The proportion of patients treated with statins was higher among men than among women (58% vs. 53%). Among both non-treated and treated patients, women were older, had a longer diabetes duration, higher BMI, more often had an estimated glomerular filtration rate (eGFR) ≤60 mL/min/1.73 m^2^, were more often treated with ≥5 chronic medications, and were less often smokers than men (Table 1). Treatment with any glucose-lowering medication was similar for men and women. A similar proportion of men and women not treated with statins had a history of CVD, whereas for those treated with statins, men were more likely than women to have a history of CVD. Men were also more often treated with a high-intensity statin than women. In the highest age groups, women appeared to have a longer diabetes duration (Appendix A). In both sexes, BMI was lower with higher age, whereas blood pressure, eGFR, and albuminuria were more unfavorable with higher age. Polypharmacy was most common in elderly women (Appendix A). The percentage of patients with statin treatment was highest in the age groups between 55 and 79 years, and lowest in, particularly, the younger as well as the oldest women (Appendix A).

### 3.1. Low-Density Lipoprotein Cholesterol, High-Density Lipoprotein Cholesterol, and Triglycerides

In patients not treated with a statin, the mean BMI-adjusted LDL-c levels were above 3 mmol/L in all age groups except in men aged ≥85 years (Figure 2A, left panel). Women had significantly lower LDL-c levels than men up to the age of 45 years and significantly higher LDL-c levels after the age of 50 years (Figure 2A left panel; Appendix A).

In patients treated with a statin, the mean BMI-adjusted LDL-c levels were below 2.5 mmol/L in all age groups (Figure 2A, right panel). There were no significant differences in LDL-c levels between men and women up to the age of 55 years. Between the age 55 and 84 years, we observed significantly higher LDL-c levels in women than in men (Figure 2A right panel; Appendix A).

Women had significantly higher BMI-adjusted HDL-c levels than men across all age groups, independent of statin treatment (Figure 2B, Appendix A).

Men had significantly higher BMI- and HbA1c-adjusted TG levels than women up to the age of 55 and 60 years when not treated and treated with a statin, respectively, and similar levels thereafter (Figure 2C, Appendix A).

The sensitivity analyses in which we additionally adjusted for the statin intensity showed similar results for LDL-c (Appendix A), HDL-c (Appendix A), and TGs (Appendix A).

### 3.2. Total Cholesterol and Non-High-Density Lipoprotein Cholesterol

In patients not treated with a statin, TC and non-HDL-c levels showed similar sex and age patterns as seen for the LDL-c levels, with lower levels in women than in men younger than 45 and 50 years, respectively, and higher levels in women than in men after the age of 50 and 55 years, respectively (Appendix A).

For those treated with a statin, women and men had similar TC levels up to the age of 50 years, but women had higher levels than men thereafter (Appendix A, Appendix A). Non-HDL levels were higher in men than in women aged 45–49 years and higher in women than in men older than 60 years, but they were similar in both sexes in other age groups (Appendix A, Appendix A).

## 4. Discussion

This study showed that differences in lipid levels between women and men with T2D change substantially across the life span. For patients not treated with a statin, women had lower LDL-c levels than men before the age of 45 years and higher LDL-c levels after the age of 50 years. Statin treatment lowered LDL-c levels in both women and men, but women still had higher LDL-c levels than men after the age of 55 years. HDL-c levels were consistently higher in women than in men in all age groups, regardless of statin treatment. TG levels were higher in men than in women before the age of 60 years, regardless of statin treatment.

### 4.1. Comparison with Existing Literature

Sex differences in LDL-c levels in T2D have previously been reported, with women having higher levels than men [16,22,26]. These studies did not allow for conclusions regarding age-dependent effects. Several other studies incorporated age in the analysis but divided the patients in only two age groups, using a cut-off of 60 or 65 years [21,24], or used broad age groups [32]. In addition, these studies were limited by not stratifying the patients by statin use. Our study adds to this knowledge, showing that higher LDL-c levels in women than men occur only after the age of 50 and 55 years among T2D patients without or with statin treatment, respectively, which is around the mean age of menopause in the Netherlands [33]. In line with our results, a previous study among 8775 T2D patients not stratifying for statin treatment found higher LDL-c levels in women than in men only after the age of 45 years [25]. In contrast, a small study of 110 patients with T2D and 74 controls did not observe sex differences in LDL-c levels between diabetic pre- and postmenopausal women [17]. This study, however, was limited by including only eight premenopausal diabetic women and did not stratify or adjust for statin treatment. Our findings show that unfavorable lipid profiles in women with T2D are particularly a postmenopausal phenomenon [10,11,34,35]. Although sex differences in LDL-c levels have been acknowledged in the general population [2,7,11,36], this is the first study presenting a detailed analysis of the differences across age groups in a large cohort of patients with T2D. We observed similar differences with respect to non-HDL-c, a proposed atherogenic lipid risk marker for patients with T2D and non-diabetic individuals [37,38,39]. The unfavorable lipid profile in women is not fully mitigated by statin treatment, since even with statin treatment, LDL-c and non-HDL-c levels in women remained higher than in men after the age of 55 years. This could be due to less intensive treatment in women [40,41], but the relationship between sex differences in treatment intensity and menopause has not been explored. We conducted a sensitivity analysis adjusting for statin intensity and observed a similar pattern of higher LDL-c levels among women after menopause. Alternative explanations for these differences need further study by considering both possible sex- (biology and physiology) as well as gender- (behavior and psychology) related differences. To reach similar LDL-c levels, women with T2D above the age of 50 may have to be treated more aggressively than men. Although, on average, T2D patients treated with a statin achieved a level of LDL-c < 2.4 mmol/L, around half of the women between 55 and 75 years of age treated with statins showed higher LDL-c levels. Particularly, in patients with T2D and additional risk factors, lower LDL-c levels might be more appropriate [18,19,42]. Our study illustrates that in women with T2D before menopause, there might still be a protective biological effect, which was previously assumed to be abrogated by the presence of T2D [17].

Previous studies looking at sex-related differences in HDL-c levels in patients with T2D reported higher HDL-c levels in women than in men [22,24,32], unrelated to patients’ age [21,24]. In line with these observations, we observed higher HDL-c levels in women in all age groups. Statin treatment did not affect HDL-c levels in our study, which is consistent with the mode of action of statins and previous research [42] and did not affect sex differences in HDL-c levels.

The high TG levels in younger and middle-aged men compared to women and older men have been observed previously in both the general population [7,43] and a population with T2D [25]. Our findings add the observation that these high TG levels in younger men are not much lowered when patients are treated with statins. This observation could be explained by a higher BMI and more visceral fat in men [44,45] but, since our analyses were adjusted for BMI, this is an unlikely explanation. The higher TG levels in young men with T2D deserve further study, particularly since the combination of increased TG levels and low HDL-c levels has been associated with a 44% increase in the occurrence of major cardiovascular events also in patients with T2D [46].

### 4.2. Strengths and Limitations

A strength of our study is the use of real-world data from a large cohort of patients with T2D, treated in primary care. To the best of our knowledge, it is the first study to investigate sex-related differences in lipid levels in these patients at different ages, with and without statin treatment. Our study also has some limitations. First, this is a cross-sectional design, so there can be potential historical demographic, nutritional, and healthcare system differences between older and younger patients included in our study. Also, mostly Caucasian people were included, which limits the application of the results to other races. Further, smoking could not be included as a confounder in the analyses due to the high proportion of missing data. Also, information on alcohol consumption and other lifestyle behaviors was not available in our database. Since such behaviors can differ between sexes and with age, this may have influenced the results due to their effects on lipid levels. In addition, we could not adjust for potential differences between men and women regarding adherence to statins. Finally, there was no information about the start of menopause in the GIANTT database, but the mean age of menopause in the Netherlands has been estimated to be 50.4 ± 4.1 years [33].

## 5. Conclusions

Among younger patients with T2D, women seemed to have a more favorable lipid profile than men, since they had lower LDL-c and TG levels and higher HDL-c levels. Younger men with T2D had particularly high TG levels. Among patients with T2D older than 50 years, women had higher LDL-c levels than men. Statin treatment partly lowered the observed sex differences, but more than half of the patients with T2D were not treated with statins. When managing cardiovascular risk in patients with T2D, more attention is needed for the menopausal status of women and for TG levels in younger men.

## Figures and Tables

**Figure 1 jcm-10-01775-f001:**
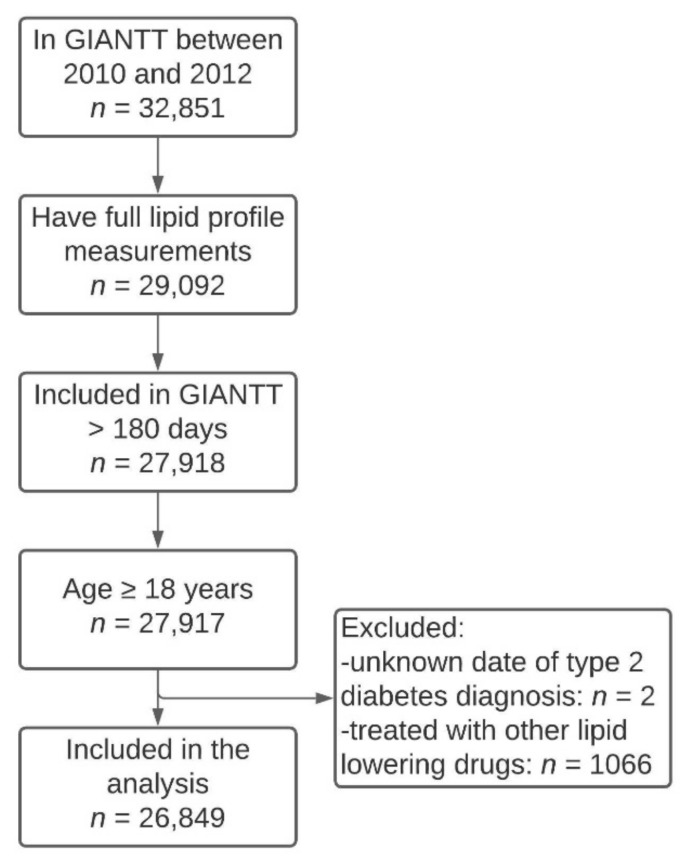
Flow chart with applied inclusion and exclusion criteria.

**Figure 2 jcm-10-01775-f002:**
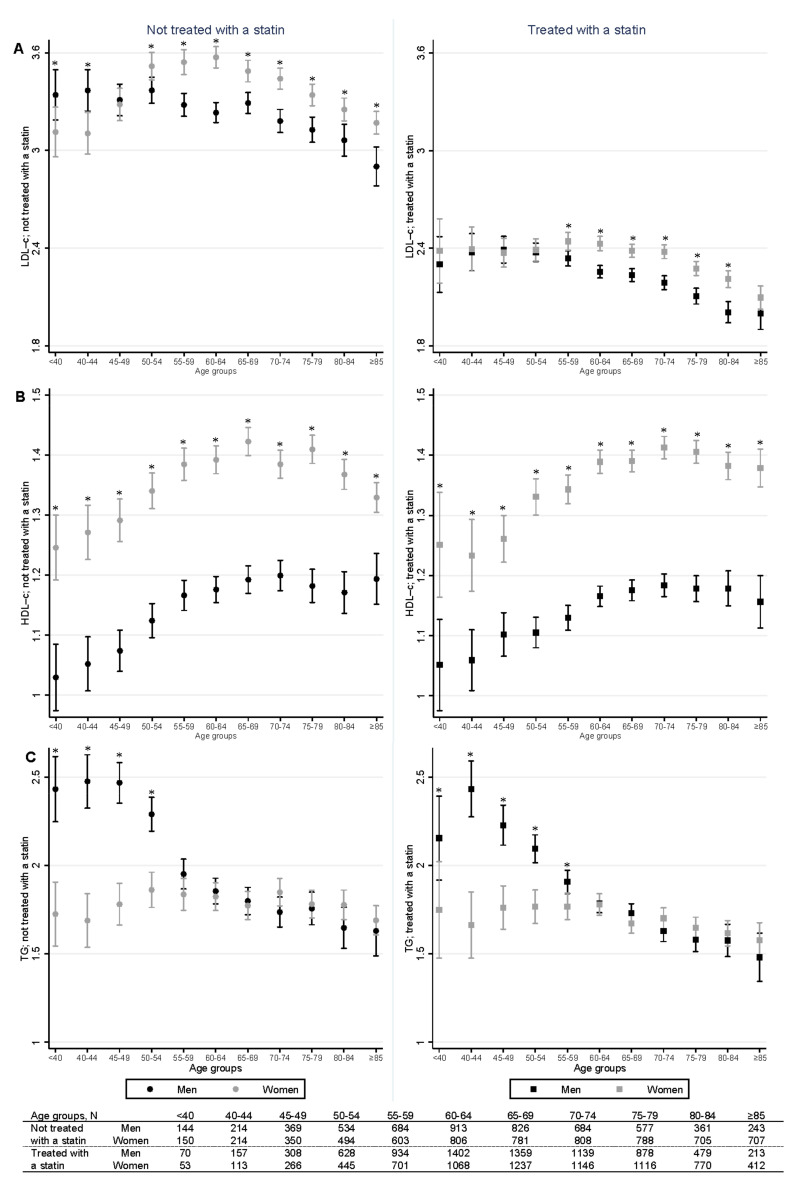
Mean lipid levels with 95% CIs for men and women per age group not treated (**left**) and treated with a statin (**right**). (**A**) Low-density lipoprotein cholesterol (LDL-c), (**B**) high-density lipoprotein cholesterol (HDL-c), and (**C**) triglyceride (TG) levels. Cholesterol measurements are in mmol/L. All values were adjusted for body mass index; TG values were additionally adjusted for glycated hemoglobin A1c. * *p* < 0.05 between men and women.

**Table 1 jcm-10-01775-t001:** Demographics of included patients.

	Not Treated with a Statin		Treated with a Statin	
	Men	Women	Men	Women
Number (%)	5549 (21)	6406 (24)	7567 (28)	7327 (27)
Age in years; mean ± SD	64 ± 12	68 ± 14	65 ± 11	68 ± 11
Diabetes duration; median (Q1–Q3)	2.5 (0.3–6.3)	3.0 (0.5–7.5)	4.6 (1.6–8.3)	5.2 (1.9–9.5)
HbA1c in % (mmol/mol); median (Q1–Q3) ^†^	6.7 (50) (6.3–7.5)	6.7 (50) (6.3–7.3)	6.8 (51) (6.4–7.4)	6.8 (51) (6.4–7.3)
BMI in kg/m^2^; mean ± SD ^†^	29.4 ± 4.9	30.8 ± 6.1	29.8 ± 4.8	30.9 ± 5.8
SBP in mmHg; mean ± SD ^†^	142 ± 19	143 ± 19	141 ± 18	142 ± 19
eGFR ≤ 60 mL/min/1.73 m^2^; *n* (%) ^†^	487 (10)	1049 (18)	802 (12)	1232 (18)
Albuminuria; *n* (%) ^†^	82 (4)	64 (3)	146 (4)	118 (4)
Polypharmacy; *n* (%)	1361 (25)	2335 (36)	4420 (58)	4780 (65)
Glucose-lowering treatment; *n* (%)	3145 (57)	3631 (57)	5965 (79)	5769 (79)
Smoking; *n* (%) ^†^	505 (25)	461 (20)	837 (27)	630 (21)
History of CVD, *n* (%) ^×^	1014 (18)	1143 (18)	2681 (35)	2007 (27)
LDL-c in mmol/L; mean ± SD	3.2 ± 0.9	3.4 ± 1.0	2.2 ± 0.7	2.3 ± 0.8
HDL-c in mmol/L; mean ± SD	1.2 ± 0.3	1.4 ± 0.4	1.2 ± 0.3	1.4 ± 0.4
Triglycerides in mmol/L; mean ± SD	1.9 ± 1.4	1.8 ± 1.0	1.8 ± 1.2	1.7 ± 0.9
TC in mmol/L; mean ± SD	5.1 ± 1.1	5.4 ± 1.1	4.0 ± 0.9	4.3 ± 0.9
Non-HDL-c in mmol/L; mean ± SD	3.9 ± 1.1	4.0 ± 1.1	2.9 ± 0.9	3.0 ± 0.9
Total/HDL-c ratio; mean ± SD	4.6 ± 1.5	4.2 ± 1.4	3.7 ± 1.1	3.3 ± 1.0
High intensity statin; *n* (%) ^‡^	n/a	n/a	693 (9)	499 (7)

^†^ Glycated hemoglobin A1c (HbA1c): 627 (2.3%) missing values; Body mass index (BMI): 4403 (16.4%) missing values; Systolic blood pressure (SBP): 9852 (36.7%) missing values; Estimated glomerular filtration rate (eGFR): 2480 (9.2%) missing values; Albuminuria: 15,473 (57.6%) missing values; Smoking: 16,447 (61.3%) missing values. DDP-4: dipeptidylpeptidase-4; CVD: cardiovascular disease; LDL: low-density lipoprotein; HDL: high-density lipoprotein; TC: total cholesterol. ^×^ Includes any record of the presence of angina pectoris, acute myocardial infarction, transient ischemic attack, stroke, atherosclerosis, other ischemic heart diseases and peripheral arterial diseases, abdominal aortic aneurysm, percutaneous transluminal (coronary) angioplasty, and peripheral or coronary bypass before the index date. ^‡^ Daily dose of atorvastatin ≥40 mg and rosuvastatin ≥20 mg; data on dose missing for eight patients.

## Data Availability

The datasets used and/or analyzed during the current study are available from the corresponding author on reasonable request.

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
