# Peer review of "Sex Differences in Lipid Profile across the Life Span in Patients with Type 2 Diabetes: A Primary Care-Based Study"

_jcm, 2021, doi:10.3390/jcm10081775_

Round 1
Reviewer 1 Report
Dr Ambrož and colleagues have evaluated sex differences in lipid profile across the life span in patients with type 2 diabetes.
I have the following comments to the submission. Line numbers refer to figures in margin.
Introduction, page 2, lines 48-49: The paper referred to (reference #17) seems to be national Dutch guidelines. Since the manuscript has been submitted to an international journal, international guidelines published in English would be preferable. Furthermore, when I looked up the "Cardiovasculair risicomanagement" Richtlijnendatabase it seems that guidelines from 2019 are now available
Material and Methods: has the Groningen Initiative to Analyse Type-2 diabetes Treatment database resulted in any previous publications? If so, please provide references. Do we know that patients included in the database are representative, or is there any selection?
I´ve connected to the link provided http://www.giant.nl but didn´t find any information in English. A reference to previous work describing the datsabase as suggested above is therefore essential to the reader not familiar with Dutch.
How come that smoking status was missing in more than 60% of registered patients? If attention to such a highly relevant variable is this low, how do we know that the rest of the information provided (medication, dosage, BMI etc) is correct? Has the database been externally validated?
Discussion:
Perhaps I haven´t looked deeply enough into this matter, but still recall an article by Berkelmans et al (European Heart Journal 2019;40:2899–906) on CVD prediction and prevention in diabetic patients of different ages which was based on lipid data from >500,000 patients.
Most patients in that study were recruited from the National Swedish Diabetes Registry, from which lipid data in type 2 diabetes had also been published previously (Eliasson et al. Eur J Prev Cardiol 2014;21:1420-8), and linked prospectively to CHD during follow-up.
Isn´t there also some overlap with the message conveyed by another Dutch paper by de Jong et al (BMJ Open Diabetes Res Care 2020 Oct;8(1):e001365. doi: 10.1136/bmjdrc-2020-001365)?
In the light of these previous findings not refererred to by the authors of the present submission, please state clearly what new information is gained from the present cross-sectional report.
References:
Please try to find references in English accessible to all readers. References #17 and 35 do not fulfil these criteria.
Author Response
Dear reviewer,
We appreciate the opportunity for the revision of our manuscript and are thankful for the comments and suggestions. Based on your feedback, we have revised the manuscript. Below, our point-by-point response to the comments is given in italics. The changes that we have made accordingly are highlighted with track-changes in the manuscript.
Dr Ambrož and colleagues have evaluated sex differences in lipid profile across the life span in patients with type 2 diabetes.
I have the following comments to the submission. Line numbers refer to figures in margin.
Introduction, page 2, lines 48-49: The paper referred to (reference #17) seems to be national Dutch guidelines. Since the manuscript has been submitted to an international journal, international guidelines published in English would be preferable. Furthermore, when I looked up the "Cardiovasculair risicomanagement" Richtlijnendatabase it seems that guidelines from 2019 are now available.
We believe that it is important to clarify that these recommendations apply for our study setting, and thus refer to the national Dutch guideline. We referred to the prevailing guideline at the time of our study period but can also refer to the more recent guideline. We have adapted this in the manuscript. We also have added an international guideline to provide a reference for the non-Dutch readers.
Material and Methods: has the Groningen Initiative to Analyse Type-2 diabetes Treatment database resulted in any previous publications? If so, please provide references. Do we know that patients included in the database are representative, or is there any selection?
I´ve connected to the link provided http://www.giant.nl but didn´t find any information in English. A reference to previous work describing the database as suggested above is therefore essential to the reader not familiar with Dutch.
We now refer to the UMCG research facilities website (https://umcgresearch.org/facilities) which provides a short description of the GIANTT database in English. In addition, we have added a more detailed description of the GIANTT database to the methods section. The GIANTT database includes almost all regional patients with T2D (<1% opted out), so there was no selection. However, the population includes mostly Caucasian people, which we have added to the manuscript in both the methods and discussion sections (line 83 and 296). More than 50 publications have been published using the GIANTT database (https://www.giantt.nl/what-is-giantt/publications). Given your comment, we now include a reference (Vos RC et al BMJ Open Diabetes Research & Care 2020) with more details about the GIANTT database with references to other relevant GIANTT publications (lines 79 to 87). See also, next comment.
How come that smoking status was missing in more than 60% of registered patients? If attention to such a highly relevant variable is this low, how do we know that the rest of the information provided (medication, dosage, BMI etc) is correct? Has the database been externally validated?
The GIANTT data are extracted from the electronic medical record systems in primary care, and thus include what is recorded in the electronic medical records of the patients. This includes information that is (1) entered manually by the general practitioners or nurse practitioners (BMI, blood pressure values, diagnoses, smoking, etc.) and (2) automatically recorded in the system (prescription data from electronic prescribing, laboratory data from the electronic laboratory reports). The low recording of smoking status in the electronic medical records is a well-known problem (see, for example, Kohane IS et J Med Internet Res 2021; 23:e22219). In our setting, this is partly due to the fact that smoking has not been included as one of the required measurements in the Dutch protocol for the annual diabetes visit in primary care, whereas weight, HbA1c and blood pressure are included. It is only mentioned that lifestyle, including smoking, should be addressed. So, although smoking status may have been discussed during consultation it may not be recorded in the medical record.
The data extraction procedure has been validated to check for completeness, that is, whether all data from the tables in the medical record systems have been imported. Before each data import units and coding of medication and diagnostic measurements are checked and harmonized. For the study period, data have also been checked by generating feedback of aggregate and patient level data at general practice level. We have added more information about these procedures in the methods section also referring to some relevant publications (Voorham J et al Pharmacoepi Drug Safety 2010; de Vries ST et al Diabetes Care 2014; de Vries FM et al Curr Med Res Opin 2015).
Discussion:
Perhaps I haven´t looked deeply enough into this matter, but still recall an article by Berkelmans et al (European Heart Journal 2019;40:2899–906) on CVD prediction and prevention in diabetic patients of different ages which was based on lipid data from >500,000 patients. Most patients in that study were recruited from the National Swedish Diabetes Registry, from which lipid data in type 2 diabetes had also been published previously (Eliasson et al Eur J Prev Cardiol 2014;21:1420-8), and linked prospectively to CHD during follow-up. Isn´t there also some overlap with the message conveyed by another Dutch paper by de Jong et al (BMJ Open Diabetes Res Care 2020 Oct;8(1):e001365. doi: 10.1136/bmjdrc-2020-001365)?
In the light of these previous findings not refererred to by the authors of the present submission, please state clearly what new information is gained from the present cross-sectional report.
Indeed, lipid profiles and sex differences have been studied previously in patients with type 2 diabetes but as far as we know none of these studies have looked at sex differences across the life span and present data for patients with and without statin treatment.
In our introduction and discussion, we mention several publications that illustrate the current body of knowledge (references number 21, 22, 24-26 and 33). We believe that the mentioned publications using the National Swedish Diabetes Registry are not particularly relevant for our study aim since they focus on CVD prediction. The study of De Jong et al is indeed relevant and we have now added this to the introduction and the discussion (lines 61, 62, 240 and 276). The study provides detailed information about sex differences in lipids among people with type 2 diabetes, but it does not answer our question regarding differences across the life span. From their age-adjusted analysis, it is not clear whether the observed sex differences occur in all age groups or are caused by including mostly women after menopause. Furthermore, they mention that their results did not change after adjusting for medication use but they do not present data on the impact of medication use across sex and age groups.
We have tried to state more clearly how our study adds to the previous studies (lines 46 to 68, 243 to 255 and 266 to 269). In particular, the novelty of our study lies in our analysis of the influence of both sex and age on the lipid profiles in patients with type 2 diabetes, differentiating between those with and without statin treatment.
References:
Please try to find references in English accessible to all readers. References #17 and 35 do not fulfil these criteria.
As mentioned above, we have added an international reference to the manuscript.
Reviewer 2 Report
This is a well written and easy to read paper, however, its overall relevance, especially to clinical practice is unclear.
On a positive note, it was pleasing to see that the LDL levels were measured directly, rather than using calculated levels.
Most importantly, It is hard to assess the clinical relevance of this information. The relevance of focusing on isolated LDL cholesterol in the aetiology of cardiovascular disease has been under debate for some time. Work by Gordon et al (1977) (10.1016/0002-9343(77)90874-9 ) demonstrates that elevated LDL alone does not significantly increase the risk of CVD. While Ravnskov et al (doi.org/10.1080/17512433.2018.1519391) argues that LDL is not causal for CVD, especially in women. This makes it harder to justify the use of statins, especially in women, particularly in light of Yourman et al's metaanalysis. (
Author Response
Dear reviewer,
We appreciate the opportunity for the revision of our manuscript and are thankful for the comments and suggestions. Based on your feedback, we have revised the manuscript. Below, our point-by-point response to the comments is given in italics. The changes that we have made accordingly are highlighted with track-changes in the manuscript.
This is a well written and easy to read paper, however, its overall relevance, especially to clinical practice is unclear.
On a positive note, it was pleasing to see that the LDL levels were measured directly, rather than using calculated levels.
Most importantly, it is hard to assess the clinical relevance of this information. The relevance of focusing on isolated LDL cholesterol in the etiology of cardiovascular disease has been under debate for some time. Work by Gordon et al (1977) (10.1016/0002-9343(77)90874-9) demonstrates that elevated LDL alone does not significantly increase the risk of CVD. While Ravnskov et al (doi.org/10.1080/17512433.2018.1519391) argues that LDL is not causal for CVD, especially in women. This makes it harder to justify the use of statins, especially in women, particularly in light of Yourman et al's metaanalysis. (doi:10.1001/jamainternmed.2020.6084). They concluded that 100 people needed to be treated for at least 2.5 years to prevent one major coronary event, with no evidence of a mortality benefit. I would like to see a stronger argument to justify increasing statin use in women if there is no apparent improvement in women's CVD risk and/or mortality.
The paper under review does not mention whether they were using statins for primary or secondary prevention, this may be an important link. Line 130 says "A similar proportion of men and women not treated with statins had a history of CVD". While this is listed in Table 1, the definition of CVD is missing. It would be clearer if the population is stratified into primary and secondary prevention.
The relevance for clinical practice relates to the guideline recommendations for patients with T2D. In general, evidence-based guidelines recommend statin use for most patients with T2D due to their increased risk for CVD, both with and without a history of CVD. In particular, the Cholesterol Treatment Trialists’ Collaborators have shown that the proportional reduction in major vascular events per mmol/L of LDL-c reduction with statins is similar between men and women with T2D. Given your comments, we now refer to this publication in the introduction (lines 56 to 58). According to practice guidelines, LDL-c levels are leading for adjusting the statin dose in clinical practice. Also high intensity statins are nowadays recommended for people with a history of CVD based on their ability to reduce the LDL-c levels to a larger extent. We therefore believe that it is of clinical relevance to look at sex and age differences in lipid profiles in relation to statin treatment in this ‘real world’ cohort. Instead of stratifying into primary and secondary prevention, we have conducted a sensitivity analysis adjusting for statin intensity. This is considered more appropriate since the dose of statin is the potential confounder for the outcomes of our study. Following your comment below, we now differentiate between moderate intensity (currently recommended for primary prevention) and high intensity (recommended for secondary prevention).
The definition of CVD was provided in the Supplementary table 1 and includes any record of the presence of angina pectoris, acute myocardial infarction, transient ischemic attack, stroke, atherosclerosis, other ischemic heart diseases and peripheral arterial diseases, abdominal aortic aneurysm, percutaneous transluminal (coronary) angioplasty, and peripheral or coronary bypass before index date. We have now added the definition also to the footnotes of Table 1.
Smoking status is an important confounder that does not appear to be well communicated. Line 96 states that smoking status was missing in more than 60% of patients and was therefore not included in the analyses. However, it is included in the demographics (table 1), but it is unclear to what these numbers relate and how representative they are of the population. Perhaps it would be better to remove completely and just leave it as a limitation.
We understand your concerns about the smoking data presented in Table 1 but feel they are still of relevance as background information for the men and women included in our study. We provided the absolute numbers, so it is clear that these are incomplete data and mention the number of missing values in the footnote of the table. We expect they reflect overestimations since smoking status is more likely to be documented when present.
The justification of the low-mod and mod-high groupings of the statins should be better justified. Both the cited reference and Vimalananda (2013) (10.1007/s11606-013-2340-5) have a low, moderate and high groups. The cited reference is ambiguous as to whether 10 or 20 mg atorvastatin or simvastatin 20-40mg should be considered moderate (Table 5). Although the reference text suggests the lower levels, should still be considered 'moderate', current clinical practice would suggests these doses would now be considered low-moderate. It would be interesting to see the changing to these groups would have a significant impact on the analysis.
Regarding the groupings of statins, we previously decided to use 2 instead of 3 groups to avoid small numbers in the low dose group, which led to this somewhat ambiguous grouping. Given your comments, we decided to redo the analysis using the groupings that are nowadays recommended for primary and secondary prevention (lines 147-149). This did not change our findings.
Lines 124/5 are ambiguous do you mean that 55% of women (51% of sample) were treated with a statin?
Thank you for pointing out this unclarity. These numbers are in relation to the whole population. We have adjusted the manuscript to make it clearer (lines 154 and 155).
Figure 2, I'm not sure the data could be better displayed by combining the "not treated with a statin" and "treated with a statin" on the same graph. Currently it is harder to compare genders across the two graphs. It is also unclear why the y-axis is broken.
We have tried combining the graphs but found that it was clearer to compare men and women in these figures separated by statin treatment. This also fits better with the aim of our study to compare the levels of men and women in those treated or not treated with a statin. The y-axis used is the same for the left and right panels for those not treated and treated with statins. The y-axis starts below the lowest levels of the confidence intervals since there is not really a zero value. We have replaced Figure 2 since a part of the y axis in the Figure 2B was missing by mistake.
It is becoming more increasing recognised that triglycerides in combination with HDL are more important risk factors for cardiovasular disease than LDL (Nichols 2018 https://doi.org/10.1111/dom.13537). Statins may help to lower TG levels. Perhaps including a more detailed analysis on the effects of statins and TG in this cohort would be beneficial.
Thank you for the suggestion. Unfortunately, the aim and design of the current study do not allow for a more detailed effect analysis of statins on TG. We do mention the need for further studies focusing on TG levels in our discussion.
Thank you for presenting a very well written paper.
We appreciate this comment, thank you.
Round 2
Reviewer 1 Report
Dear colleagues, thanks for answering my comments and questions and revising your manuscript accordingly. Best regards.
Reviewer 2 Report
Thank you for addressing the issues